# Predicting Atrial Fibrillation Treatment Outcome with Siamese Multi-modal Fusion and Cardiac Digital Twins

**Alexander M Zolotarev** [1,3]                    A.ZOLOTAREV@QMUL.AC.UK
**Abbas Khan Rayabat Khan** [2,3]                  A.RAYABATKHAN@QMUL.AC.UK
**Gregory Slabaugh** [2,3]                         G.SLABAUGH@QMUL.AC.UK
**Caroline H Roney** [1,3]                         C.RONEY@QMUL.AC.UK

[1] *School of Engineering and Materials Science, Queen Mary University of London, UK*

[2] *School of Electronic Engineering and Computer Science, Queen Mary University of London, UK*

[3] *Queen Mary's Digital Environment Research Institute (DERI), London, UK*

**Editors:** Accepted for publication at MIDL 2024

## Abstract

Atrial fibrillation, the most common heart rhythm disorder, presents challenges in treatment due to difficulty pinpointing the patient-specific regions of abnormal electrical activity. While biophysical simulations of cardiac electrophysiology create a digital twin of atrial electrical activity based on CT or MRI scans, testing various treatment strategies on them is time-consuming and impractical on clinical timescales. Our proposed pipeline, incorporating Siamese architecture, fuses latent representations of multi-modal features extracted from atrial digital twin before any therapy and predicts the outcomes of several treatment strategies. A large in-silico dataset of 1000 virtual patients, generated from clinical data, was utilized to provide the biophysical simulations before (used for feature extraction) and after (used for calculating ground truth labels depending on whether atrial fibrillation terminates or not) various treatment strategies. By accurately predicting freedom from atrial fibrillation, our pipeline paves the way for personalized atrial fibrillation therapy with a fast and precise selection of optimal treatments. The code is available at https://github.com/pcmlab/AF_ablation_DT.git.

**Keywords:** Electrophysiology, Digital twins, Multi-modal fusion, Personalized medicine

## 1. Introduction

Atrial fibrillation (AF) remains the most common heart rhythm disease and is highly associated with an increased risk of stroke (Joglar et al., 2024). Multiple clinical and experimental studies showed that AF may be caused and maintained by specific localized sources of pathological electrical activity (Hansen et al., 2015). The locations of AF sources and their conduction pathways are highly dependent on the arrhythmogenic substrate, including fibrotic tissue distribution (Hansen et al., 2017). Fibrosis is remodelled heart tissue with specific conduction parameters. It plays a crucial role in AF maintenance and can be used as a feature map to predict AF treatment outcome (Althoff et al., 2022).

The treatment of AF may be surgical and consists of ablation (physical destruction) of tissue areas with pathological sources of electrical activity. The baseline ablation strategy recommended by clinical guidelines is pulmonary vein isolation (PVI), which uses radiofrequency catheter ablation to electrically isolate the pulmonary veins from the rest of the heart, preventing triggers from the pulmonary veins from maintaining AF (Joglar et al.,

2024). However, PVI alone is usually not enough to successfully eliminate AF, and the long-term success rate for persistent AF ablation therapy is between 50 and 60%. There are multiple options for additional ablation lesions to stop the propagation of pathological electrical activity through the atria, including linear ablations, isolation of the left atrial appendage, or ablation of low-voltage areas and fibrosis (Rosa et al., 2021).

Human digital twins allow assessment of patient-specific features and simulation of physiological processes. They can provide personalised treatments incorporating information on both patient-specific anatomy and physiology. Specifically, the digital twin of the human heart can help to predict sudden cardiac death and to guide the optimal treatment for heart rhythm disorders (Trayanova et al., 2023) such as AF using biophysical simulations.

However, AF biophysical models require a long simulation time, and predicting the long-term response to any therapy is challenging. Merging AF simulations with Deep Learning (DL) algorithms can overcome this limitation and allow us to predict the outcomes of various ablation strategies on clinical timescales. These predictions can base on the patient-specific characteristics of the electrical activity within the heart - by analyzing the atrial shape, its structure and parameters of electrical signals (anatomical and physiological features). As an example, fibrosis distributions are unique for each patient and affect the patterns of electrical activity. Late Gadolinium Enhancement (LGE) MRI is a non-invasive tool for estimating fibrosis quantitatively. Gadolinium accumulates in fibrotic tissue in higher amounts than in healthy tissue or the blood pool, and as a result, the intensity of LGE-MRI images from scar tissue is higher. Image intensity ratio (IIR, MRI intensity divided by the intensity of the blood pool) provides a method for quantifying atrial fibrosis distribution, standardized across different patients and MRI machines (Khurram et al., 2014).

Dominant Frequency (DF) and Phase Singularity (PS) maps have been proposed to be relevant attributes to characterize atrial electrical activity. Spectral analysis of atrial signals can locate the regions with the highest frequencies likely to maintain arrhythmia. In particular, DF maps visualize the frequency corresponding to the highest peak in the Fourier spectrum for each spatial location. Ablation of areas of high DF results in AF termination (Sanders et al., 2005) and the frequency features, including DF, may serve as successful indicators for localizing the area of AF by ML classifiers (Zolotarev et al., 2020). As a further metric, the phase of transmembrane voltages can detect regions with rotational patterns which may sustain fibrillation. Specifically, phase singularities (which indicate the centres of rotational activity and areas of wavefront breakup) were shown to serve as a marker for the drivers which maintain AF pattern (Umapathy et al., 2010).

In this study, we propose a novel deep learning approach to predict the outcomes (the probability of atrial fibrillation termination) of different treatment strategies based on a large in-silico study of 1000 virtual patients. Our paper makes the following contributions:

- We introduce a novel deep learning pipeline based on a Siamese architecture that fuses multi-modal feature maps derived from both atria.

- We produce the methodology of generating multi-modal feature maps using an atrial digital twin before any treatment.

- We show the effectiveness of the proposed pipeline and demonstrate that the full combination of multi-modal feature maps, Siamese architecture and fusion strategy is important to achieve the predictive capability.

## 2. Methods

**Dataset** We used 1000 bi-atrial meshes from a statistical shape model derived from Cardiac CT scans of 19 healthy patients (Rodero et al., 2021). In brief, right (RA) and left (LA) atrium were segmented from CT scans, and the surface meshes were utilized to form the bi-layer model of the heart. To create a more realistic representation of cardiac tissue, different anatomical structures were added to the mesh, such as inter-atrial connections to imitate the signal propagation between the atria, the crista terminalis and pectinate muscles, and the sinoatrial node, which is the heart's primary pacemaker (Roney et al., 2023). We also mapped the fibre field to the atrial surfaces from the same bi-layer atlas to include heterogeneity within the tissue by highlighting the primary direction of signal propagation with the highest velocity and the perpendicular direction with a slower speed.

To assign these structures, we utilised the *universal atrial coordinates (UAC)* (Roney et al., 2019). *UAC* were calculated for each mesh by solving a Laplace equation with boundary conditions. This allows the comparison of spatial data across patients and the construction of a 2D representation of any 3D feature map. Next, each bi-atrial mesh was assigned two (for LA and RA) randomly selected fibrosis distributions from 100 clinical LGE-MRI scans of 100 AF (43 paroxysmal and 57 persistent) patients undergoing first ablation, as described in (Roney et al., 2022). Based on the fibrosis distributions, the ionic and conductivity properties of fibrotic regions were changed to represent atrial remodelling. All of the aforementioned resulted in creating the in-silico dataset of 1000 virtual patients, which can serve as an initial foundation for biophysical simulations (Figure 1A).

The biophysical simulations were performed by solving the monodomain equation (Eq. 1, (Clayton et al., 2011) ) using openCARP software with the Courtemanche ionic model for the cellular action potential (Courtemanche et al., 1998):

$$\frac{\partial V}{\partial t} = \nabla \cdot \tilde{\mathbf{D}} \nabla V - \frac{I_{\text{ion}}}{C_{\text{m}}}, \tag{1}$$

where $V$ is transmembrane potential, $I_{\text{ion}}$ - transmembrane ionic current per unit area, $C_m$ - the membrane capacitance per unit area, and $\tilde{\mathbf{D}}$ - diffusion tensor $(m^2 \ s^{-1})$. AF was initiated by adding four Archimedean spiral waves on the atrial surface, with more details at (Roney et al., 2020). AF episodes were simulated for 15 seconds to have enough information for phase and frequency analysis and feature extraction.

Four types of ablation strategies were applied after 5 seconds of AF simulation across the virtual cohort, including PVI and PVI with LA, RA and bi-atrial fibrosis ablation. We applied ablation lesions by setting the tissue conductivity to 0.001 S/m for the corresponding elements of the atrial mesh (Bayer et al., 2016). PVI ablations were simulated by adding two non-conducting rings around the left and right pulmonary vein antra.

An automatic methodology was utilised to ablate the regions with higher fibrosis intensity. Firstly, IIR distributions for LGE maps were binarised with the threshold of 1.22 (Roney et al., 2022). Regions of positive labels were connected to create several clusters of high fibrosis. Finally, each cluster was connected with non-conductive lesions to either PVI lesions or to heart valve borders (which are also non-conductive), depending on which was closer. The resulting lesion maze prevented the pathological electrical signals from freely propagating through the atrial surface and may lead to AF termination.

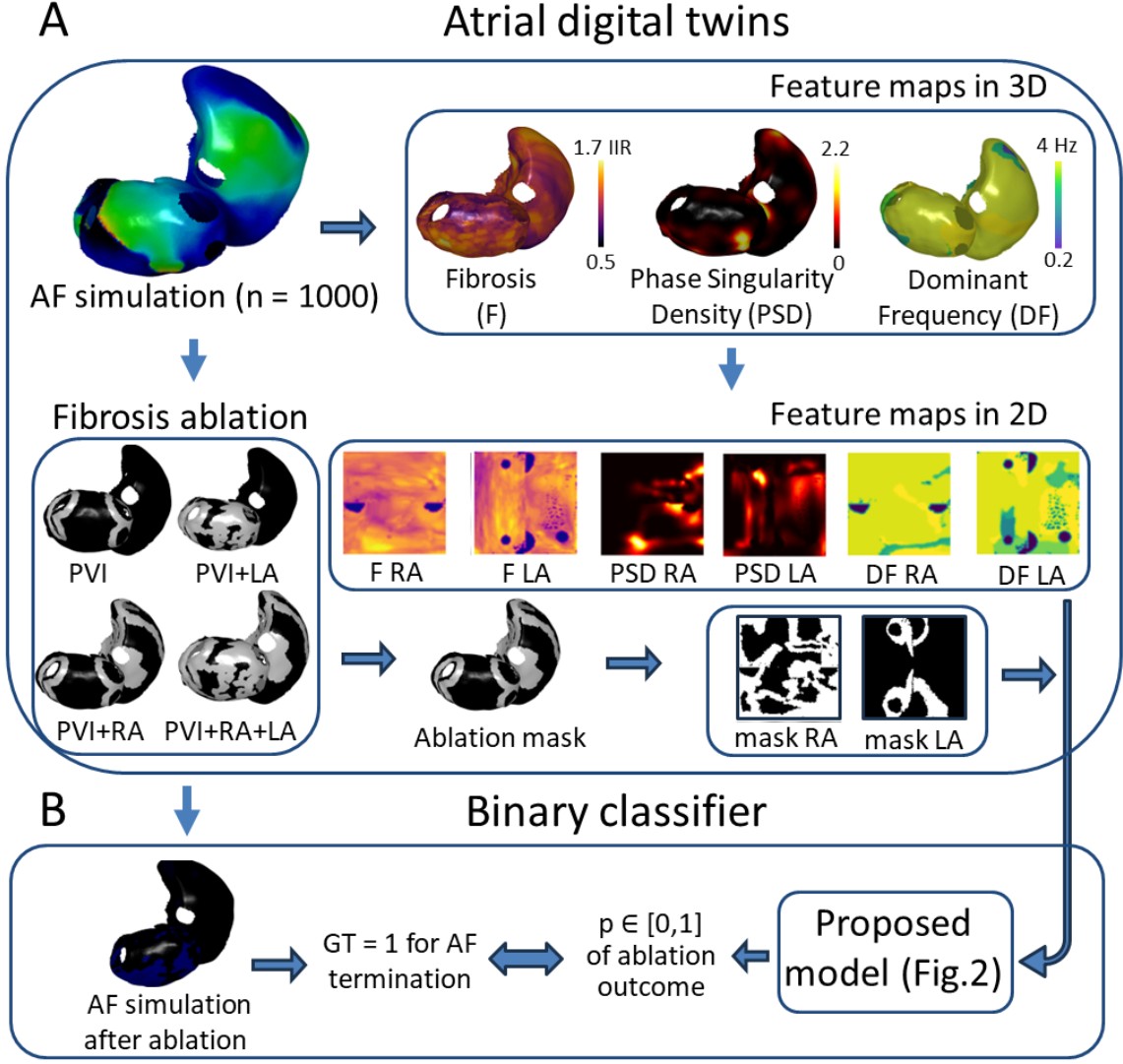

Figure 1: Overview of our pipeline for binary prediction of Atrial Fibrillation (AF) ablation outcome. A - Atrial digital twin based on AF simulation before any ablation provides the feature maps for prediction. B - The proposed model utilises feature maps and the patient-specific ablation mask to predict the probability of AF termination after ablation, which can be checked by analyzing the AF simulation after ablation. PVI - Pulmonary Vein Isolation, LA, RA - Left/Right Atrium, GT - ground truth label, IIR - Image Intensity Ratio.

**Feature maps** The DL pipeline was trained based on different input feature maps, including an ablation mask for each type of ablation and additional anatomical (fibrosis distribution), as well as physiological (Phase Singularity Density, PSD, and Dominant

Frequency, DF) features calculated from the AF simulations before any ablation. AF simulations yield transmembrane voltage recordings with a temporal resolution of 50 ms for all nodes on the atrial mesh. The DF values for each mesh element were calculated as the frequency of the highest spectral peak of the signal, excluding peaks of the spectrum above 20 Hz to remove physiologically uninformative frequencies from the analysis. The phase of atrial signals was calculated by applying the Hilbert transform after subtracting the signal mean. We constructed PSD maps by counting the number of PS occurrences (by calculating the topological charge (Rogers, 2004)) across the whole duration of the AF simulation at each mesh element and smoothing the values through the mesh.

For both PSD and DF values on a 3D atrial mesh, we used the $UAC$ to convert them from 3D to 2D feature maps. Initially, we set up the shape of 2D maps, which was selected as 128 by 128 pixels, with a normalized coordinate system (0-1) on both axes. We calculated the 2D $UAC$ coordinates for each mesh element on LA and RA surfaces, where both the $x$ and $y$ axes also span between 0 and 1. Each pixel of the 2D map corresponds with a mesh element with a minimal distance between the pixel centre and the 2D location of this mesh element. By assigning DF and PSD values from selected mesh elements to corresponding 2D pixels for both LA and RA, we generated four 2D feature maps: DF LA, DF RA, PSD LA, PSD RA. Finally, we created 2D ablation masks and fibrosis maps based on the same protocol as for DF and PSD maps (Figure 1A). Labels of ablation lesion elements from the 3D atrial mesh served as the basis for ablation masks (mask LA, mask RA), while fibrosis distributions from LGE-MRI scans were used to create fibrosis feature maps (F LA, F RA).

**Deep Learning pipeline and implementation** The binary outcomes of simulations after different ablation strategies were used for training the DL pipeline and were assigned a ground truth (GT) label, whether AF terminates or not. We counted the AF episode as terminated if there is no electrical activity at the end of the recording (specifically, the last peak of the action potential is within the range 0 - 60% of the whole duration of the recording). To define the final GT, we calculated the last peak's values of the AF signal for each node of both LA and RA and averaged the value across all nodes (Figure 1B).

To address the problem of fast and correct prediction of AF ablation, we propose a DL pipeline based on Siamese architecture (Figure 2). Each head consists of DenseNet121 (Huang et al., 2016) network and utilises 4 channel-wisely concatenated feature maps (ablation mask, PSD, DF and fibrosis maps, all of the size 128 by 128 pixels) from LA and RA separately. The outputs of both heads (n = 32 each) were fed into a Multi-modal Outer Arithmetic Block (MOAB) (Alwazzan et al., 2023) for the fusion of latent representations of features. It allows us to capture and combine the relevant features from different anatomical structures by applying four arithmetic operations. The resulting arrays were concatenated into multi-modal tensor which passed through the 2D convolution layer. The final prediction of AF ablation outcome as the probability of AF termination was achieved through two successive fully-connected layers (n = 1089 and 512 respectively) and one dropout layer.

The performance was evaluated by accuracy, ROC-AUC and F1-score on a testing set of 250 virtual patients. To avoid data leakage, we assigned the virtual patients with the same fibrosis distribution randomly selected from 100 LGE-MRI maps to either the training or testing set. The training was conducted on an NVIDIA GeForce RTX 3080 video card using Binary Cross Entropy loss and Adam optimizer with a learning rate of 4e-7.

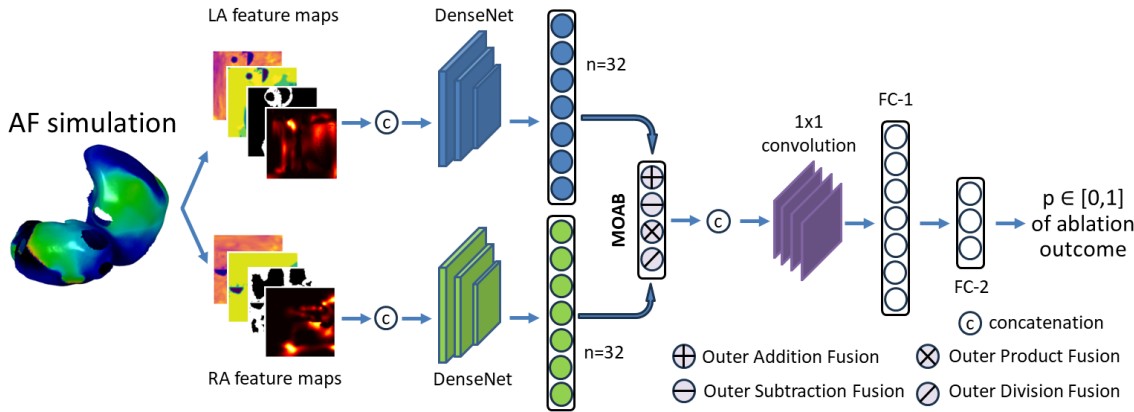

Figure 2: Flowchart of DL pipeline. LA, RA - Left/Right Atrium, MOAB - Multi-modal Outer Arithmetic Block, FC - Fully-Connected layer.

## 3. Results

We show the effectiveness of the proposed model on the dataset of 1000 virtual patients described in Section 2. For each virtual patient, we ran the biophysical simulations with AF initialization before and after 4 types of ablation strategies. All 1000 AF simulations before ablation were sustained (AF was stable during the whole duration of recording). Overall, the percentage of successful ablations (which lead to AF termination) was 41%. Analysing this by ablation therapy approach, the effectiveness of PVI ablation was 25%, PVI with LA fibrosis ablation - 36%, PVI with RA fibrosis ablation - 39%, and PVI with bi-atrial fibrosis ablation - 60%. An example of one virtual patient with only PVI with bi-atrial fibrosis ablation resulting in AF termination is shown in Figure 3.

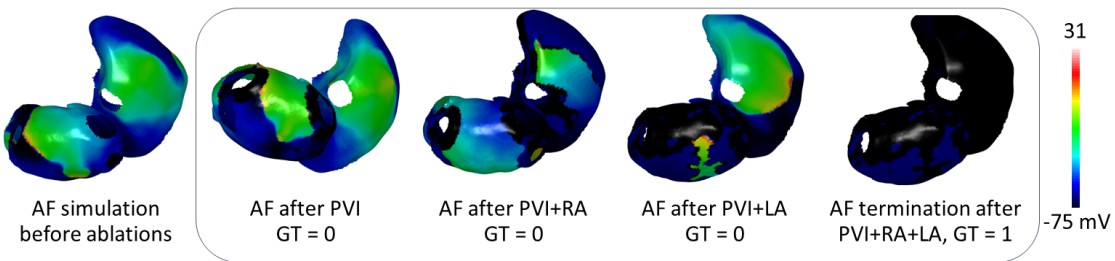

Figure 3: The snapshots of electrical activity during 5 AF simulations for one patient. PVI - Pulmonary Vein Isolation, LA, RA - Left/Right Atrium, GT - Ground Truth.

We demonstrated that the proposed model achieved high performance in predicting the outcome of different ablation strategies, including a ROC-AUC of 0.92, an accuracy of 87%, and an F1-score of 0.88. In particular, the outcomes of PVI ablation alone are the

| Methods | ROC-AUC | Accuracy | F1-score |
|---|---|---|---|
| Single head | 0.88 | 0.86 | **0.88** |
| Two heads + concatenation | 0.90 | 0.85 | 0.87 |
| EfficientNet backbone + MOAB fusion | 0.90 | 0.85 | 0.86 |
| Two heads for all vs masks features + MOAB fusion | 0.89 | 0.84 | 0.85 |
| Two heads for LA vs RA features + MOAB fusion | **0.92** | **0.87** | **0.88** |

Table 1: Quantitative results for ablation studies.

easiest to predict, with ROC-AUC of 0.93 and 21 incorrectly predicted cases (out of 250). In contrast, 44 cases of PVI with RA fibrosis ablation were predicted incorrectly, which resulted in a ROC-AUC of 0.904 for this type of ablation. Specifically, the confusion matrices for predictions in general and for each ablation type are provided in Appendix A. The running time for one ablation strategy prediction was 41 seconds (34 - for the feature extraction and 7 - for the inference), which means that the DL pipeline can predict AF treatment outcome more than 3 times faster in comparison with conventional biophysical simulation (the fastest model run was at least 134 seconds on the UK national supercomputer ARCHER2).

We conducted an extensive ablation study to test the individual effects of each component of the proposed DL pipeline (Table 1). Firstly, we checked whether it is necessary to use two heads by comparing it to the model's performance if we used a one-headed pipeline (all feature maps (n=8) were initially concatenated and fed into the DenseNet121 network to provide the final prediction after passing fully-connected layer and softmax activation). Then, we tested what would happen if we changed only one feature from the best implementation (two DenseNet121 heads with MOAB fusion of outputs). We utilised the different backbone architecture for two heads (EfficientNet) rather than DenseNet121, then the simple concatenation of outputs rather than MOAB fusion, and have shown that the original model has the best metrics. Finally, we showed that separating feature maps before training into LA and RA parts (which is physiologically relevant) performs better than separating them into 2 ablation masks vs all (fibrosis, PSD and DF maps) feature maps.

We tried to answer the question of how different input feature maps affect the performance of the proposed model. For comparison, we selected the best-performing implementation consisting of two DenseNet121 heads with input features from the left and right atrium and outputs fused by MOAB. We found that the baseline model trained solely on the ablation mask acquired a ROC-AUC of 0.69, whereas incorporating fibrosis, DF or PSD elevated the metric to 0.86, 0.87, and 0.91, respectively. Furthermore, combining all features led to a ROC-AUC of 0.92 (Table 2).

| Input features | ROC-AUC | Accuracy | F1-score |
|---|---|---|---|
| Mask only | 0.69 | 0.70 | 0.75 |
| Fibrosis + mask | 0.86 | 0.80 | 0.83 |
| DF + mask | 0.87 | 0.82 | 0.84 |
| PSD + mask | 0.91 | 0.85 | 0.87 |
| DF + PSD + fibrosis + mask | **0.92** | **0.87** | **0.88** |

Table 2: The effect of input features on the performance.

**Discussion** A recent multi-centre clinical study showed that ablating fibrosis in left atrium (LA) after PVI does not change the rate of AF recurrence in comparison with PVI alone (Marrouche et al., 2022). However, this study investigated the effect of fibrosis ablation targeting the LA only, while it has been shown by an experimental study of adenosine-related AF (Li et al., 2016) that fibrotic tissue in right atrium (RA) can also serve as an anchor for AF pathways. Therefore, RA fibrosis areas could serve as an ablation target, and we have shown that ablating these areas improves the outcomes in a large in-silico trial. We have previously demonstrated that machine learning classifiers can predict long-term AF recurrence rates for 100 AF patients with high accuracy. Also, we found that ROC-AUC values varied from 0.61 if relying only on patient history to 0.85 if adding imaging and simulated physiological inputs (Roney et al., 2022). Confirming these findings we have currently shown that selecting input features is crucial and adding physiological characteristics such as PSD improves the results.

**Limitations** Our study is not without limitations. Firstly, all findings are based on synthetic data and further validation on clinical recordings is needed. Secondly, all atrial meshes were mapped by the same fibre field and the effect of adding different fibre maps on the same atrial mesh can be our future task. The area near the sinoatrial node was not excluded from the RA ablation masks. 1000 meshes were generated from a limited number of atrial anatomies, and although it was shown that the anatomical volumes of this virtual cohort were within the range of values from UK Biobank cohort of over 5000 individuals (Petersen et al., 2017), the generalizability issue should be tested more carefully on a bigger set of clinical meshes. The threshold for fibrosis distributions and the resolution of feature maps were set up to be 1.22 and 128 respectively, and changing the values of these hyperparameters may affect the results. The current simulation data used for training our pipeline assumes homogeneous electrophysiological properties in tissue conductivities used across healthy or fibrotic tissue. However, fibrotic remodelling is multifactorial, which may affect AF properties (Roney et al., 2016). Lastly, fibrosis distributions were randomly assigned to each of the LA and RA separately meaning that virtual patients may have a LA fibrosis distribution from a paroxysmal patient and RA from a persistent one, or any such combination. Our future work will investigate the effects of AF classification on fibrosis distribution and simulated AF as well as the effects of type of remodelling on prediction.

## 4. Conclusion

This study presents a novel approach for predicting the outcome of atrial fibrillation ablation by integrating an atrial digital twin with a deep learning pipeline. The proposed pipeline leverages a Siamese architecture with multi-modal fusion of latent representations, resulting in superior performance. Furthermore, by incorporating anatomical and physiological features from pre-ablation atrial fibrillation simulations into the model, we demonstrate significant improvement in predictive capability compared to solely using an ablation mask.

## Acknowledgments

We acknowledge ARCHER2 simulation funding, the openCARP software team, the support from the NIHR Barts BRC (NIHR203330) and the UKRI FLF (MR/W004720/1) grants.

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

## Appendix A.  Confusion matrices

| Simulated | Predicted | |
|---|---|---|
| | AF termination | AF |
| AF termination | 365 | 67 |
| AF | 70 | 498 |

Table 3: Confusion matrix for testing the proposed pipeline on 250 virtual patients.

| Simulated | Predicted PVI | | Predicted PVI with RA fibrosis | | Predicted PVI with LA fibrosis | | Predicted PVI with LA and RA fibrosis | |
|---|---|---|---|---|---|---|---|---|
| | AF termination | AF | AF termination | AF | AF termination | AF | AF termination | AF |
| AF termination | 57 | 13 | 86 | 23 | 78 | 18 | 144 | 13 |
| AF | 8 | 172 | 21 | 120 | 15 | 139 | 26 | 67 |

Table 4: Detailed confusion matrices for each ablation type.

