# OpenReview forum: "Predicting Atrial Fibrillation Treatment Outcome with Siamese Multi-modal Fusion and Cardiac Digital Twins"
_MIDL.io/2024/Conference — MIDL 2024 Oral_

### Official Review · Reviewer_67uK · 2024-02-25

**Confidence:** 5
**Preliminary Rating:** 4
**Recommendation:** Poster
**Final Rating:** 5

**Summary:**

The authors present a novel framework to predict the outcome of ablation therapy to treat atrial fibrillation (AF). Specifically, a digital twin of the left and right atrium is incorporated to provide electrophysiological features and training data for the deep-learning-based classifier. For each the left and right atrium, the dominant frequency and phase singularity map are derived from AF simulations before ablation and combined with a fibrosis and an ablation map to form the features for a Siamese network build on DenseNet. The proposed pipeline was evaluated in two ablation studies by training it on 750 geometries and evaluating it on 250 unseen geometries. On this synthetic dataset, the proposed method demonstrates high accuracy in classifying whether the AF terminates under a specific ablation pattern.

**Strengths:**

- The paper is well written and the topic of predicting the outcome of AF ablation is valuable to the community
- The method is novel in that it smartly combines information from an atrial electrophysiological digital twin with structural features and by fusing features from the left and right atrium
- The method is extensively evaluated and the results are encouraging, despite using only synthetic data for training and evaluating the method

**Weaknesses:**

- The proposed pipeline is only trained and evaluated on synthetic data

- The training data seems to include only simulations with fixed electrophysiological parameters (properties of the ionic model and conductivity of fibrotic and healthy tissue), which may not reflect the inter-patient tissue variability

- In both the method and results section, the paper misses several details to fully understand and assess the contribution (see detailed comments)

- The paper lacks a critical discussion of the results

**Detailed Comments:**

- Introduction: While all statements in the introduction are correct, I believe that it would greatly help newcomers to the field to add few more citations that support the statements, e.g., please add to the statement "PVI is the baseline ablation strategy recommended by clinicial guidelines" a citation to one of these guidelines.

- Equation 1: Shouldn't the diffusion term be divided by C_m? If I recall it correctly, the monodomain equation is also missing the variable to denote the membrane surface-to-volume ratio. Please double-check the equation.

- Section 3.1, last paragraph: Could the authors please comment on whether such a lesion maze is clinically plausible?

- Training / test data: Could the authors please confirm (or clarify) that the 1000 synthetic geometries were split into 75% training and 25% testing? Was an additional validation set used to fine-tune the hyperparameters? How were the shapes sampled from the shape model? Were any special considerations taken to sample the test shapes?

- Evaluation: Since the effectiveness of the ablation is varying significantly between PVI only and PVI + biatrial fibrosis ablation, it would be very valuable to understand to what degree this imbalance impacts the results, e.g., by showing confusion matrices for each ablation type.

- One argument for the proposed pipeline is a significantly faster selection of the optimal ablation strategy compared to running several AF simulations with varying ablation protocols. Could the authors please comment on the run-time of the proposed method? How long does the inference of the DL network take in comparison to running the "baseline" AF simulation, i.e., before ablation, and extracting the electrophysiological features from the simulation?

- Table 1: Could the authors please provide more information on the ablation experiments? Was special care taken so that each method has approximately the same number of parameters and an adequate latent dimension size before the fully-connected prediction head? On the experiment to separate the ablation masks vs all other masks: Was this network also trained as a Siamese network? If it was trained as a Siamese network, how was the input dimension of the 2 ablation masks adapted to match the 6-dimensional input of the other head? In addition, how can this be justified since I would assume that each head must learn significantly different features?

- Table 2: The results of using PSD + mask are very close to using all features. Could the authors please comment on whether the improvement is statistically significant?

**Justification Of Final Rating:**

The authors have clarified all questions and concerns in great detail (big thank you to the authors!). The changes made to the manuscript are appropriate and further boost the quality of the already good manuscript. In addition to general improvements in text, the proper discussion of the results as well as the additional insights through the confusion matrices greatly improve the paper. I am confident that this submission will be of great interest to the community.

**Justification Of The Preliminary Rating:**

I strongly believe that the proposed method has merit and that it would be of value to the MIDL community, in particular because it successfully fuses an atrial digital twin with structural features and DL. However, there are several weaknesses to be addressed. Most importantly, a critical discussion of the results should be added since it would greatly strengthen the paper.

**Questions To Address In The Rebuttal:**

In addition to the points raised under weaknesses and in the detailed comments, could the authors please comment on what role the resolution of the raw data, the mesh, and the feature maps might play onto the results?

**Special Issue:**

Yes

---

> ### Author Response · Authors · 2024-03-17
>
> (#3.1) The proposed pipeline is only trained and evaluated on synthetic data
>
> We thank the reviewer for this comment. By using synthetic data, we have more control over the impact of different feature maps including fibrosis distributions, we were able to test different ablation approaches, and it enabled us to choose and validate the optimal configuration of a Multi-modal Fusion approach. At a later stage we wish to validate our approach on clinical data.
>
> (#3.2) The training data seems to include only simulations with fixed electrophysiological parameters (properties of the ionic model and conductivity of fibrotic and healthy tissue), which may not reflect the inter-patient tissue variability
>
> We thank the reviewer for this question. In this study, we utilised different distributions of fibrosis to capture different distributions of conduction velocity and effective refractory period in our simulations. Our future work will investigate the effects of different types of fibrotic remodelling and restitution heterogeneity on our findings and whether the addition of these heterogeneities require additional input features. This will open new research directions such as what spatial resolution of restitution data is required for prediction. We have added the following to the limitations of our paper: “The current simulation data used for training our pipeline assumes homogeneous electrophysiological properties in tissue conductivities used across healthy or fibrotic tissue. However, fibrotic remodelling is multifactorial, which may affect AF properties (https://pubmed.ncbi.nlm.nih.gov/28011842/). Our future work will investigate the effects of AF classification on fibrosis distribution and simulated AF as well as the effects of type of remodelling on prediction.”
>
> (#3.3) In both the method and results section, the paper misses several details to fully understand and assess the contribution (see detailed comments)
>
> Thank you for noticing these omissions, we have replied to each of comments below.
>
> (#3.4) The paper lacks a critical discussion of the results
>
> We thank the reviewer for this comment and have added the separate discussion section to our manuscript.
>
> (#3.5) Introduction: While all statements in the introduction are correct, I believe that it would greatly help newcomers to the field to add few more citations that support the statements, e.g., please add to the statement "PVI is the baseline ablation strategy recommended by clinicial guidelines" a citation to one of these guidelines.
>
> We agree with the reviewer and add the reference to the most recent guideline, Joglar 2023, section 8.4.2.
>
> (#3.6) Equation 1: Shouldn't the diffusion term be divided by C_m? If I recall it correctly, the monodomain equation is also missing the variable to denote the membrane surface-to-volume ratio. Please double-check the equation.
>
> We used the form proposed in Clayton 2010 paper, equation 10 (https://doi.org/10.1016/j.pbiomolbio.2010.05.008). For more clarification, we have added this reference to the sentence with the equation and the units of diffusion tensor (m^(2) s^(-1)).
>
> (#3.7) Section 3.1, last paragraph: Could the authors please comment on whether such a lesion maze is clinically plausible?
>
> Yes, the ablation lesion set for either PVI or PVI plus any fibrosis ablation is clinically relevant. Specifically, we refer to the DECAAF II clinical trial, where the operator either encircled or covered with ablation lesions all fibrotic areas (10.1001/jama.2022.8831, section Fibrosis-Guided Ablation). However, we need to state that these ablation lesion sets cannot be directly applied to clinical patients. For example, we did not consider the ease of ablating different regions and we did not exclude lesions occurring in the vicinity of the sinoatrial node, the primary pacemaker of the heart, even though a clinician would be careful to not ablate these regions. We have added this information to the Limitations section: “The area near the sinoatrial node was not excluded from the RA ablation masks.”.
>
> (#3.8) Training / test data: Could the authors please confirm (or clarify) that the 1000 synthetic geometries were split into 75% training and 25% testing? Was an additional validation set used to fine-tune the hyperparameters? How were the shapes sampled from the shape model? Were any special considerations taken to sample the test shapes?
>
> Yes, we confirm that 1000 synthetic geometries were split into 75% training and 25% testing set, we didn’t fine-tune the hyperparameters on the additional validation set. The cases for the testing set were selected following the rule that a given fibrosis distribution can be within either the training or testing set.

---

> > ### Author Response · Authors · 2024-03-17
> >
> > (#3.9) Evaluation: Since the effectiveness of the ablation is varying significantly between PVI only and PVI + biatrial fibrosis ablation, it would be very valuable to understand to what degree this imbalance impacts the results, e.g., by showing confusion matrices for each ablation type.
> >
> > We thank the reviewer for this comment and want to highlight that we have not currently separated the training set into 4 datasets of different ablation strategies (all of them are unbalanced). The training set has a combination of all ablation strategies and has a good balance between positive and negative values (41:59). We found that confusion matrices for each ablation strategy will be helpful for the readers and have added them to the Appendix (as well as the general confusion matrix with respect to Q#1.1).
> >
> > (#3.10) One argument for the proposed pipeline is a significantly faster selection of the optimal ablation strategy compared to running several AF simulations with varying ablation protocols. Could the authors please comment on the run-time of the proposed method? How long does the inference of the DL network take in comparison to running the "baseline" AF simulation, i.e., before ablation, and extracting the electrophysiological features from the simulation?
> >
> > We thank the reviewer for pointing out this important information. To answer this question, we compare the running times of biophysical simulation and DL prediction for one hold-out case, which was PVI with bi-atrial fibrosis ablation for one virtual patient. The running time of AF ablation simulations was 351 seconds while running on the local computer with 64 core desktop CPUs and 134 seconds on the UK national supercomputer ARCHER2. In contrast, the running time for Deep Learning prediction was 41 seconds (34 seconds for extracting the feature maps and 7 seconds for the inference on NVIDIA GeForce RTX 3080 video card). Therefore, our proposed model can predict the ablation outcome from 3 to 8 times faster than conventional biophysical simulation, depending on the power of the CPU for simulations.  We have added this information to the results: “The running time for one ablation strategy prediction was 41 seconds (34 seconds for the feature extraction and 7 seconds for the inference), which means that the DL pipeline can predict AF treatment outcome more than 3 times faster in comparison with conventional biophysical simulations (the fastest model run was at least 134 seconds on the UK national supercomputer ARCHER2).”.
> >
> > (#3.11) Table 1: Could the authors please provide more information on the ablation experiments? Was special care taken so that each method has approximately the same number of parameters and an adequate latent dimension size before the fully-connected prediction head? On the experiment to separate the ablation masks vs all other masks: Was this network also trained as a Siamese network? If it was trained as a Siamese network, how was the input dimension of the 2 ablation masks adapted to match the 6-dimensional input of the other head? In addition, how can this be justified since I would assume that each head must learn significantly different features?
> >
> > Thank you for this interesting question. We have not currently tested any modifications of the network architecture related to ablation studies. During ablation masks vs all feature maps the network was trained in a Siamese (but not symmetric) mode, with 2 input channels for masks and 6 channels for feature maps (in contrast with 4 channels for the left and 4 channels for the right atrium as a default). The number of output features after passing the DenseNet121 remained the same as in the original MOAB paper (n=32) - for both fusion between 4 and 4 channels and the fusion between 2 and 6 channels.
> >
> > (#3.12) Table 2: The results of using PSD + mask are very close to using all features. Could the authors please comment on whether the improvement is statistically significant?
> >
> > To respond to this question, we compare the metrics on 4-fold cross-validation between the full set of input feature maps and PSD maps and ablation masks only. The resulting ROC-AUCs are shown in the Table below. The student t-test indicates that there is no statistically significant difference between these two sets of metrics (p-value = 0.08).
> >
> > full  vs PSD + mask:
> >
> > 1st fold - 0.917 vs 0.914
> > 2nd fold - 0.904 vs 0.876
> > 3rd fold - 0.933 vs 0.907
> > 4th fold - 0.863 vs 0.856

---

> > > ### Author Response · Authors · 2024-03-17
> > >
> > > (#3.13) In addition to the points raised under weaknesses and in the detailed comments, could the authors please comment on what role the resolution of the raw data, the mesh, and the feature maps might play onto the results?
> > >
> > > We thank the reviewer for this valuable comment. The resolution significantly affects the results. Specifically, we did an additional ablation study to test how model performance depends on the resolution of the feature maps. We found that ROC-AUC scores were 0.908, 0.903, and 0.917 for resolutions of 64, 96, and 128 pixels. We did not test the effect of mesh resolution on the performance on 1000 SSM but another study from our group has shown that downsampling of the mesh used for analysis from the high-resolution simulation mesh can lead to a large change in phase singularities distributions (https://www.touchcardio.com/imaging/journal-articles/27-spatial-resolution-requirements-for-analysing-atrial-fibrillation-wavefront-patterns-insights-from-virtual-patient-models/). However, further investigations are needed to determine the optimal balance between data resolution, prediction metrics and run times. We have added this information to the Limitation section: “The threshold for fibrosis distributions and the resolution of feature maps were set up to be 1.22 and 128 respectively, and changing the values of these hyperparameters may affect the results.”).

---

### Official Review · Reviewer_UzzR · 2024-02-28

**Confidence:** 4
**Preliminary Rating:** 3
**Recommendation:** Poster
**Final Rating:** 4

**Summary:**

This paper presents a Siamese network for fusing multiple modalities of data, including those output from digital simulations, to predict the outcome to 4 different A-fib treatment. The input to the model includes PSD, DF, and fibrosis maps along with the ablation mask representing ablation strategies. The output is a binary prediction of whether the ablation is successful. Experiments are conducted on virtual patients, showing the ability of the model to predict outcome in held-out patients.

**Strengths:**

The proposed Siamese network training is based on an extensive set of simulation data — the effort is commendable. The fusion of functional data (PSD and DF) and structural data is interesting. The goal to estimate the outcome of an ablation, instead of simple prediction of disease state, is also interesting.

The results obtained on virtual patients are encouraging.

**Weaknesses:**

The lack of any indication of the clinical validity of the prediction is a main weakness of the present study.

The sample size in terms of the unique patients contributing to the basis of the virtual patient cohort is somewhat limited — 19 healthy patients. The 1000 bi-atrial meshes are obtained by combining these meshes with 100 different fibrosis distributions. How would this limit the variability of the simulation and thus the generalizability of the subsequent outcome prediction model is not clear.

**Detailed Comments:**

Understandably clinical validation on this type of setup is difficult — nevertheless, on the actual patients that contribute atrial mesh or fibrosis map to the virtual cohort, there must be history of ablation treatment and outcome. Can this be used as retrospective validation of the proposed prediction network, at least as a proof of concept.

While the virtual patient is left out based on their fibrosis distribution, what about the atrial mesh?

**Justification Of Final Rating:**

The authors addressed some of my questions. Given the challenging nature of clinical validation for this type of works, and the extensive amount of works put forth in the current work, I'm happy to recommend the acceptance of this work. It could raise good discussion within the MIDL community regarding how digital twins and AI can benefit each other.

**Justification Of The Preliminary Rating:**

The proposed work is overall interesting and represents significant work built on sophisticated simulation pipeline. Some major lingering concerns have to do with the indications of clinical validity of the prediction network and the potential limitation due to limited samples size, although this may be acceptable as a preliminary proof of concept.

**Questions To Address In The Rebuttal:**

It’d be helpful if the authors can respond to the questions about the possible clinical validation no matter how limited, as well as how the split of virtual patients is determined based on the atrial mesh used; comments are generalizability due to the limited unique atrial geometry and fibrosis samples should also be added.

**Special Issue:**

No

---

> ### Author Response · Authors · 2024-03-17
>
> (#2.1) Understandably clinical validation on this type of setup is difficult — nevertheless, on the actual patients that contribute atrial mesh or fibrosis map to the virtual cohort, there must be history of ablation treatment and outcome. Can this be used as retrospective validation of the proposed prediction network, at least as a proof of concept.
>
> The subjects that were used for constructing the anatomical statistical shape model do not have any history of AF treatment as they are healthy asymptomatic patients who went to the emergency with acute chest pain and had no cardiac conditions during follow-up. To incorporate fibrosis in the models, we used the LGE data of 100 patients undergoing ablation therapy for AF from a previous study (10.1161/CIRCEP.121.010253) where only left atrial simulations were performed. In our current study, we decided to use bi-atrial simulations to include the effects of the right atrium and right atrial fibrosis. We are currently working on constructing bi-atrial versions of these 100 cases for clinical validation. Please see more discussion about it in the response to Q#2.3 where we also point to new text in the manuscript.
>
> (#2.2) While the virtual patient is left out based on their fibrosis distribution, what about the atrial mesh?
>
> Please see our response to the previous question. Briefly, we had only left atrial (LA) segmentations and could perform AF biophysical simulations only on LA meshes; however, we are working on constructing bi-atrial versions of these 100 cases for clinical validation.
>
> (#2.3) The sample size in terms of the unique patients contributing to the basis of the virtual patient cohort is somewhat limited — 19 healthy patients. The 1000 bi-atrial meshes are obtained by combining these meshes with 100 different fibrosis distributions. How would this limit the variability of the simulation and thus the generalizability of the subsequent outcome prediction model is not clear. It’d be helpful if the authors can respond to the questions about the possible clinical validation no matter how limited, as well as how the split of virtual patients is determined based on the atrial mesh used; comments are generalizability due to the limited unique atrial geometry and fibrosis samples should also be added.
>
> The reviewer indicates a very important aspect of our work on how to translate these findings into clinical practice or at least show its clinical validity. Our ultimate goal is clinical validation of the proposed pipeline, and we present in this paper the first step of developing our approach on simulated data. We hope that the pipeline even without clinical validation can be interesting for data science and cardiac modelling communities and can serve as a proof-of-concept example of merging a novel deep learning architecture and a large in-silico trial of cardiac digital twins with different AF treatment strategies. By using synthetic data, we have more control over the impact of different feature maps including fibrosis distributions, we were able to test different ablation approaches, and it enabled us to choose and validate the optimal configuration of a Multi-modal Fusion approach.
>
> We are currently working on extensive clinical validation by adding a separate hold-out testing set of clinical patients as well as creating a more representative second version of the Statistical Shape Models based on additional CT/MRI scans for cases with AF. However, this process is time-consuming and requires significant development; as such, we unfortunately cannot perform new AF simulations for testing now. Once we have completed the bi-atrial segmentation of these cases with known ablation approaches and outcomes, we will test our deep learning pipeline predictions against the known outcome.
>
> Focusing on the generalizability issue, we want to emphasize that the authors of the original paper with 1000 Statistical Shape Models have shown that the anatomical volumes of the CT cohort were within the range of values from the UK Bio-bank in a population cohort of over 5000 individuals [https://doi.org/10.1186/s12968-017-0327-9 ]. Regarding the concerns of clinical validation and generalizability, we have added the following sentences to limitations section: “Firstly, all findings are based on synthetic data and further validation on clinical recordings is needed. 1000 meshes were generated from a limited number of atrial anatomies, and although it was shown that the anatomical volumes of this virtual cohort were within the range of values from UK Biobank in a population cohort of over 5000 individuals, the generalizability issue should be tested more carefully using a bigger set of clinical meshes.”.

---

> > ### Comment · Reviewer_UzzR · 2024-03-19
> > **Follow-up question**
> >
> > Thanks for the clarification.
> >
> > Outside the issue of clinical validation, can the authors clarify if the training/test split considered leaving out some LA geometry? It was not clear in the response above.

---

> > > ### Author Response · Authors · 2024-03-20
> > >
> > > Hi, if I figure out the question correctly,
> > > 1000 synthetic geometries were split into 75% training and 25% testing sets, which means that 250 virtual patients in the testing set are based on 250 unique SSM geometries. All virtual patients are bi-atrial models based on 1000 SSM and 100 LGE distributions from LA. We did not use these real LA geometries anywhere, because we do not have RA geometries for them to construct bi-atrial models.

---

> > > > ### Comment · Reviewer_UzzR · 2024-03-20
> > > >
> > > > Thanks for the clarification.
> > > >
> > > > Although more extensive and clinical validation are required in the future works, I appreciate the extensive amount of works involved in this proof of concept and acknowledge the difficulty in clinical validation. I'm happy to recommend the acceptance of this MIDL paper and look forward to seeing the future iterations of this interesting work.

---

### Official Review · Reviewer_PLBx · 2024-02-29

**Confidence:** 4
**Preliminary Rating:** 3
**Recommendation:** Poster
**Final Rating:** 4

**Summary:**

The authors present a method to assess the success of treatment ablation based on feature maps coming from imaging data and biophysical simulations using a digital twin created from CT or MRI scans. The long-term response of the treatment is predicted using a siamese neural network without the need of running long simulation times.

**Strengths:**

- Achieves a good prediction of the treatment response using as input features obtained form biophysical simulations and imaging data.

- Highlights the importance of fibrosis maps in the treatment planning for AFib vs traditional approaches.

**Weaknesses:**

- In this in-silico dataset the success rates of the tested ablation procedures are in the range [25%-60%]. Thus, the best procedure including LA and RA fibrosis maps achieves the success rate of PVI in clinical practice. Therefore, it seems that it could be easier to bias the network to predict mostly not successful ablation, specially in the PVI case for the in-silico data. Would it make sense to balance the training between successful and not successful cases? Would it be possible to include a confusion matrix for the predictions?

**Detailed Comments:**

- What’s the balance between paroxismal and persistent maps assigned to the in-silico data?

- Future steps or limitations of the current work in the conclusion would be appreciated.

- The authors used a fixed threshold for the fibrosis maps. Nevertheless, to my understanding, the fibrosis quantification from LGE is semi-quantitative, i.e: the values are scaled with respect to healthy tissue. Thus, subjects with high levels of fibrosis would not have a good 'healthy' reference and the amount of fibrotic tissue might not be estimated correctly. If that's the case, how could this threshold impact the results?

**Justification Of Final Rating:**

The authors have clarified all my doubts, and my concerns were properly addressed. Furthermore, they did a great effort updating the paper and providing further evidence as well as detailed comments. I think it is a good paper, and I am confident it will be of interest for the community.

**Justification Of The Preliminary Rating:**

The paper is interesting, it shows the potential of personalised approaches to improve treatment outcome and how it can be combined with deep learning approaches to speed-up certain aspects of the treatment planning. Nevertheless, there are some aspects of the methodology from which I would like to know a bit more details.

**Questions To Address In The Rebuttal:**

- The fiber directions are mapped from a reference atlas. Thus, I assume that most of the subjects present a similar distribution. What is the effect of fibers direction? , I also assume that they play an important role in the conduction dynamics. From Table 2, PSD seems to be the most relevant feature map for the prediction, while fibrosis not so much, might it be related to the fibers direction? i.e: maybe the network is learning which 'fibers' ablate to better stop the abnormal patterns.

- In Section 3.1 the authors state that they sample 2 fibrosis distributions, on for each atrium, do these distributions come from the same sampled subject?

**Special Issue:**

No

---

> ### Author Response · Authors · 2024-03-17
>
> R#1
>
> (#1.1) In this in-silico dataset the success rates of the tested ablation procedures are in the range [25%-60%]. Thus, the best procedure including LA and RA fibrosis maps achieves the success rate of PVI in clinical practice. Therefore, it seems that it could be easier to bias the network to predict mostly not successful ablation, specially in the PVI case for the in-silico data. Would it make sense to balance the training between successful and not successful cases? Would it be possible to include a confusion matrix for the predictions?
>
> We thank the reviewer for this valuable comment. The proportion of positive and negative values in the overall dataset is 41:59 (first paragraph of Results), which means that our dataset is well-balanced. We would like to indicate that we trained the model on all ablation options simultaneously and did not aim to separate datasets based on the ablation type. We also agree that adding the stratification into the dataloader for training is a good idea in the case of unbalanced datasets. This will become important if we look at the different ablation approaches separately (e.g. in the case of PVI ablation alone). We also decided to add to the Appendix a table with the confusion matrix for the prediction and link to this Table to the main manuscript.
>
> (#1.2) What’s the balance between paroxismal and persistent maps assigned to the in-silico data?
>
> The balance between fibrosis distributions from paroxysmal and persistent AF patients is 43 to 57 cases, as stated in section 3.1. However, we did not create the virtual patients based on only paroxysmal or persistent maps, these distributions were assigned to each of the LA and RA randomly to generate virtual patients with a range of fibrosis distributions. As such, the virtual patients may have a fibrosis distribution in the LA from a paroxysmal case, and in the RA from a persistent case, or indeed any combination. This choice means that the models capture different degrees and distributions of fibrosis observed across the spectrum of AF patients, but we cannot label virtual patients as paroxysmal or persistent. The effect of the AF classification (paroxysmal or persistent) on fibrosis distributions and resulting AF simulations is an interesting question and can be a task for investigation in future research. We have added the following to our limitations and future work: "Lastly, fibrosis distributions were randomly assigned to each of the LA and RA separately meaning that virtual patients may have a LA fibrosis distribution from a paroxysmal patient and RA from a persistent one, or any such combination. Our future work will investigate the effects of AF classification on fibrosis distribution and simulated AF.”
>
> (#1.3) Future steps or limitations of the current work in the conclusion would be appreciated.
>
> We appreciate this comment and have added the relevant information to the manuscript section of limitations.
>
> (#1.4) The authors used a fixed threshold for the fibrosis maps. Nevertheless, to my understanding, the fibrosis quantification from LGE is semi-quantitative, i.e: the values are scaled with respect to healthy tissue. Thus, subjects with high levels of fibrosis would not have a good 'healthy' reference and the amount of fibrotic tissue might not be estimated correctly. If that's the case, how could this threshold impact the results?
>
> We thank the reviewer for this question. Fibrosis intensities were normalized by the mean blood pool intensity to account for differences in image contrast. This scaling is more robust and less subjective than normalizing with respect to healthy tissue. However, the selection of a threshold value for the regions in which fibrosis was incorporated in the models has a significant impact on the results. To provide more clarity we have stated the information about normalization in the manuscript (“Image intensity ratio (IIR, MRI intensity divided by the intensity of the blood pool https://doi.org/10.1016/j.hrthm.2013.10.007) provides a method for quantifying atrial fibrosis distribution, standardized across different patients and MRI machines.”). We also added the sentence about the threshold to the Limitations section (“The threshold for fibrosis distributions and the resolution of feature maps were set up to be 1.22 and 128 respectively, and changing the values of these hyperparameters may affect the results.”).

---

> > ### Author Response · Authors · 2024-03-17
> >
> > (#1.5) The fiber directions are mapped from a reference atlas. Thus, I assume that most of the subjects present a similar distribution. What is the effect of fibers direction? I also assume that they play an important role in the conduction dynamics. From Table 2, PSD seems to be the most relevant feature map for the prediction, while fibrosis not so much, might it be related to the fibers direction? i.e: maybe the network is learning which 'fibers' ablate to better stop the abnormal patterns.
> >
> > We have mapped the same fibre field from a bi-atrial bilayer atlas for all virtual patients; therefore, with the current set of simulations, fibre orientation cannot be used as a valuable feature for learning. However, fibre orientation plays a role in signal propagation and the effect of adding different fibre maps on the same atrial meshes can be our future task. To further clarify this point, we changed the sentence in the Methods section (“We also mapped the fibre field to the atrial surfaces from the same bi-layer atlas to include heterogeneity within the tissue by highlighting the primary direction of signal propagation with the highest velocity and the perpendicular direction with a slower speed.”) and added the information about it to Limitations (“All atrial meshes included the same fibre field and the effect of adding different fibre maps will be investigated in future studies .”).
> >
> > (#1.6) In Section 3.1 the authors state that they sample 2 fibrosis distributions, on for each atrium, do these distributions come from the same sampled subject?
> >
> > These distributions were randomly assigned to atrial meshes and can be either from the same or different patients. We have stated this in the Methods section: “Next, each bi-atrial mesh was assigned two (for LA and RA) randomly selected fibrosis distributions from 100 clinical LGE-MRI scans”. We checked the virtual cohort and found that 1% of all patients have fibrosis distributions on LA and RA which came from the same patient.

---

### Author Response · Authors · 2024-03-17

We thank the editor and the reviewers for their comments and questions. We have provided comprehensive answers and have changed the manuscript to address their points; specifically, we removed the Abbreviations section and added new Discussion, and Limitations sections and Appendix. Regarding the strictly limited size of the manuscript, we have also slightly changed the text; however, all findings, tables and figures remain the same.

---

### Meta-Review · Area_Chair_qGXo · 2024-04-03

**Recommendation:** Accept (Oral)
**Confidence:** 5

**Metareview:**

This paper received the ratings with 1 borderline, 1 weak accept and 1 strong accept. The authors have addressed the major concerns during the rebuttal. Therefore a decision of accept is recommended. I strongly suggest the authors take the reviewers’ comments into the final revision.

---

### Decision · Program_Chairs · 2024-04-05

Accept (Oral)